# Integration of genomics and transcriptomics predicts diabetic retinopathy susceptibility genes

Andrew D Skol[1], Segun C Jung[2], Ana Marija Sokovic[3], Siquan Chen[4], Sarah Fazal[4], Olukayode Sosina[5,6], Poulami P Borkar[3], Amy Lin[3], Maria Sverdlov[3], Dingcai Cao[3], Anand Swaroop[6], Ionut Bebu[7], DCCT/EDIC Study group, Barbara E Stranger[8]*, Michael A Grassi[3]*

[1]Department of Pathology and Laboratory Medicine, Ann and Robert H. Lurie Children's Hospital of Chicago, Chicago, United States; [2]Research and Development, NeoGenomics Laboratories, Aliso Viejo, United States; [3]University of Illinois at Chicago, Chicago, United States; [4]Cellular Screening Center, Office of Shared Research Facilities, The University of Chicago, Chicago, United States; [5]Department of Biostatistics, Johns Hopkins University, Baltimore, United States; [6]National Eye Institute, National Institutes of Health (NIH), Bethesda, United States; [7]The George Washington University, Biostatistics Center, Rockville, United States; [8]Department of Pharmacology, Center for Genetic Medicine, Northwestern University Feinberg School of Medicine, Chicago, United States

**Abstract** We determined differential gene expression in response to high glucose in lymphoblastoid cell lines derived from matched individuals with type 1 diabetes with and without retinopathy. Those genes exhibiting the largest difference in glucose response were assessed for association with diabetic retinopathy in a genome-wide association study meta-analysis. Expression quantitative trait loci (eQTLs) of the glucose response genes were tested for association with diabetic retinopathy. We detected an enrichment of the eQTLs from the glucose response genes among small association p-values and identified folliculin (*FLCN*) as a susceptibility gene for diabetic retinopathy. Expression of *FLCN* in response to glucose was greater in individuals with diabetic retinopathy. Independent cohorts of individuals with diabetes revealed an association of *FLCN* eQTLs with diabetic retinopathy. Mendelian randomization confirmed a direct positive effect of increased *FLCN* expression on retinopathy. Integrating genetic association with gene expression implicated *FLCN* as a disease gene for diabetic retinopathy.

*For correspondence:
barbara.stranger@northwestern.edu (BES);
grassim@uic.edu (MAG)

Competing interests: The authors declare that no competing interests exist.

## Introduction

Almost all individuals with diabetes will develop some form of diabetic retinopathy over time (*National Diabetes Fact Sheet, 2011*). In the United States diabetic retinopathy is the most frequent cause of blindness among working age individuals (*Centers for Disease Control and Prevention, 2018*). Interindividual variation contributes significantly to susceptibility of the severe manifestations of diabetic retinopathy, which results in vision impairment. Epidemiological studies suggest that phenotypic variation is influenced by two primary risk factors: the duration of diabetes and an individual's level of glycemia (HbA1c) (*DCCT/EDIC Research Group et al., 2017*). However, these two factors do not completely explain an individual's risk for developing diabetic retinopathy. For instance, a common anecdotal clinical experience is the comparison of patients with similar durations of diabetes and similar levels of glycemic control who have tremendously disparate clinical

**eLife digest** One of the side effects of diabetes is loss of vision from diabetic retinopathy, which is caused by injury to the light sensing tissue in the eye, the retina. Almost all individuals with diabetes develop diabetic retinopathy to some extent, and it is the leading cause of irreversible vision loss in working-age adults in the United States. How long a person has been living with diabetes, the extent of increased blood sugars and genetics all contribute to the risk and severity of diabetic retinopathy. Unfortunately, virtually no genes associated with diabetic retinopathy have yet been identified.

When a gene is activated, it produces messenger molecules known as mRNA that are used by cells as instructions to produce proteins. The analysis of mRNA molecules, as well as genes themselves, can reveal the role of certain genes in disease. The studies of all genes and their associated mRNAs are respectively called genomics and transcriptomics. Genomics reveals what genes are present, while transcriptomics shows how active genes are in different cells.

Skol et al. developed methods to study genomics and transcriptomics together to help discover genes that cause diabetic retinopathy. Genes involved in how cells respond to high blood sugar were first identified using cells grown in the lab. By comparing the activity of these genes in people with and without retinopathy the study identified genes associated with an increased risk of retinopathy in diabetes. In people with retinopathy, the activity of the folliculin gene (FLCN) increased more in response to high blood sugar. This was further verified with independent groups of people and using computer models to estimate the effect of different versions of the folliculin gene.

The methods used here could be applied to understand complex genetics in other diseases. The results provide new understanding of the effects of diabetes. They may also help in the development of new treatments for diabetic retinopathy, which are likely to improve on the current approach of using laser surgery or injections into the eye.

outcomes for diabetic retinopathy. Moreover, some individuals with diabetes develop very minimal retinopathy (*Sun et al., 2011*), whereas others clearly seem to have a predisposition for severe reti-nopathy (*Gao et al., 2014*).

Together, these observations in conjunction with the high concordance of diabetic retinopathy between family members support an underlying genetic mechanism. Familial aggregation and twin studies estimate that genetic factors account for 25–50% of an individual's risk of developing severe diabetic retinopathy (*Arar et al., 2008*; *Hietala et al., 2008*). Unfortunately, little is known about the genetic architecture that contributes to susceptibility for diabetic retinopathy. Genetic studies suggest that it is a highly polygenic trait influenced by multiple genetic variants of small effect. Our group and others have performed genome-wide association studies to better delineate the molecu-lar factors that predispose to diabetic retinopathy (*Grassi et al., 2011*; *Grassi et al., 2012*; *Pollack et al., 2019*). However, these studies have had limited success, likely due to insufficient study sample sizes and the phenotypic heterogeneity of diabetic retinopathy.

Notably, like other complex disease traits including age-related macular degeneration (*Fritsche et al., 2014*; *Fritsche et al., 2016*), a majority of genetic variants nominally associated with diabetic retinopathy are located in intronic or inter-genic regions (*Risch and Merikangas, 1996*). Most of these variants appear to play critically important functional roles in regulating gene expres-sion. In fact, several of the top associated SNPs identified in our meta-GWAS of diabetic retinopathy (*Grassi et al., 2011*) are present in DNase hypersensitivity sites and affect gene expression levels by altering the allelic chromatin state or the binding sites of transcription factors (*Maurano et al., 2012*).

The observation that disease-associated genetic loci often influence gene expression levels (*Gamazon et al., 2018*) led us to postulate that integrating gene expression with genetic association would be a powerful approach to identify susceptibility genes for diabetic retinopathy. We hypothe-sized that cell lines derived from individuals with diabetes with and without retinopathy could be used to uncover genetic variation that explain individual differences in the response to diabetes. Cul-turing two sets of cell lines under controlled, identical conditions from individuals with diabetes who

did and those who did not develop retinopathy could unmask molecular differences in how these groups respond to glucose (*Grassi et al., 2016*; *Grassi et al., 2014*). We presumed that a portion of those differences would have a genetic basis.

In this article, we identify genes whose expression responds differently to glucose in cells derived from T1D individuals with and without diabetic retinopathy. We show that one of these genes, folliculin (*FLCN)*, is causally implicated in diabetic retinopathy based on results from genetic association testing and Mendelian randomization.

## Results

### Individuals with retinopathy (PDR) show differences in diabetes duration and level of glycemia compared to individuals without retinopathy (nDR)

Matched DCCT/EDIC participants (for age, sex, treatment group, cohort, and diabetes duration) from whom the gene expression profiling was obtained are detailed in *Supplementary file 1a*. All individuals had T1D and were Caucasian, and 60% were female. As anticipated, notable differences were observed between individuals with and without retinopathy (PDR vs. nDR) for mean duration of diabetes ($53 \pm 43.4$ months vs. $27 \pm 13.4$ months). as it was also not possible to completely match participant pairs for this covariate or for level of glycemia (HbA1c).mean HbA1c ($9.71 \pm 2.37$ vs. $7.62 \pm 1.07$) given their significant impact on retinopathy.

### Interindividual variation is evident in the transcriptional response to glucose

We quantified gene expression levels from lymphoblastoid cell lines (LCLs) of all study individuals (nDM, PDR, and nDR) in both standard glucose (SG) and high glucose (HG) conditions and determined the genome-wide transcriptional response to glucose for each individual ($RG_{all}$). We observed that 22% of 11,548 examined genes demonstrated a differential response in expression between the two conditions (true positive rate; $\pi_1 = 0.22$) (*Storey and Tibshirani, 2003*; *Figure 1—figure supplement 1*), with 299 of those at an FDR < 0.05 (*Figure 1a*), supporting a significant impact of glucose on the LCL transcriptome. We confirmed that interindividual transcriptome response to HG is greater than the intraindividual response ($p=2\times10^{-16}$) (*Figure 1—figure supplement 2*, *Figure 1—figure supplement 3*, *Figure 1—figure supplement 4*). Interestingly, *TXNIP*, the most highly glucose-inducible gene in multiple cell types (*Devi et al., 2017*; *Chen et al., 2016*), exhibited the largest ($\log_2$(FC) difference = 0.2) and most significant ($p=3.2\times10^{-12}$, FDR = $5.1\times10^{-8}$) transcriptional response to glucose. Pathway analysis using gene set enrichment analysis (GSEA) revealed dramatic upregulation of genes involved in structural changes to DNA (DNA packaging, FDR < 0.0001) and in genes such as transcription factors that modulate the cellular response to environmental stimuli (protein DNA complex, FDR < 0.0001) (*Figure 1b*). Conversely, genes that modulate the cellular response to infection were considerably downregulated (type 1 interferon, FDR < 0.0001; gamma interferon, FDR < 0.0001; leukocyte chemotaxis genes, FDR < 0.0001) potentially supporting earlier work that chronic glucose exposure depresses cellular immune responsiveness (*Delamaire et al., 1997*; *Al-Mashat et al., 2006*).

### Individuals with diabetic retinopathy exhibit a differential transcriptional response to glucose

We observed differences in the transcriptional response to glucose between matched individuals with and without diabetic retinopathy ($RG_{pdr-ndr}$). Principal component analysis (PCA) demonstrated that the observed interindividual variance is dominated by randomized DCCT treatment (intensive vs. conventional) group effects based on retinopathy status ($p=3\times10^{-6}$) (*Figure 2—figure supplement 1*) and is not confounded by LCL growth rate (p>0.05) or EBV-(Epstein Barr virus) copy number (p>0.05). Using a gene-wise analysis we identified 103 genes exhibiting a differential glucose response between individuals with and without retinopathy (p<0.01) (*Figure 2*; *Supplementary file 1b*). Some of these genes and pathways have previously been shown to play a role in diabetic retinopathy. One of the top differential response genes was *IL1B* (p=0.008, $\log_2$(FC) response difference = 0.289). Expression of *IL1B* has been previously reported to be induced by

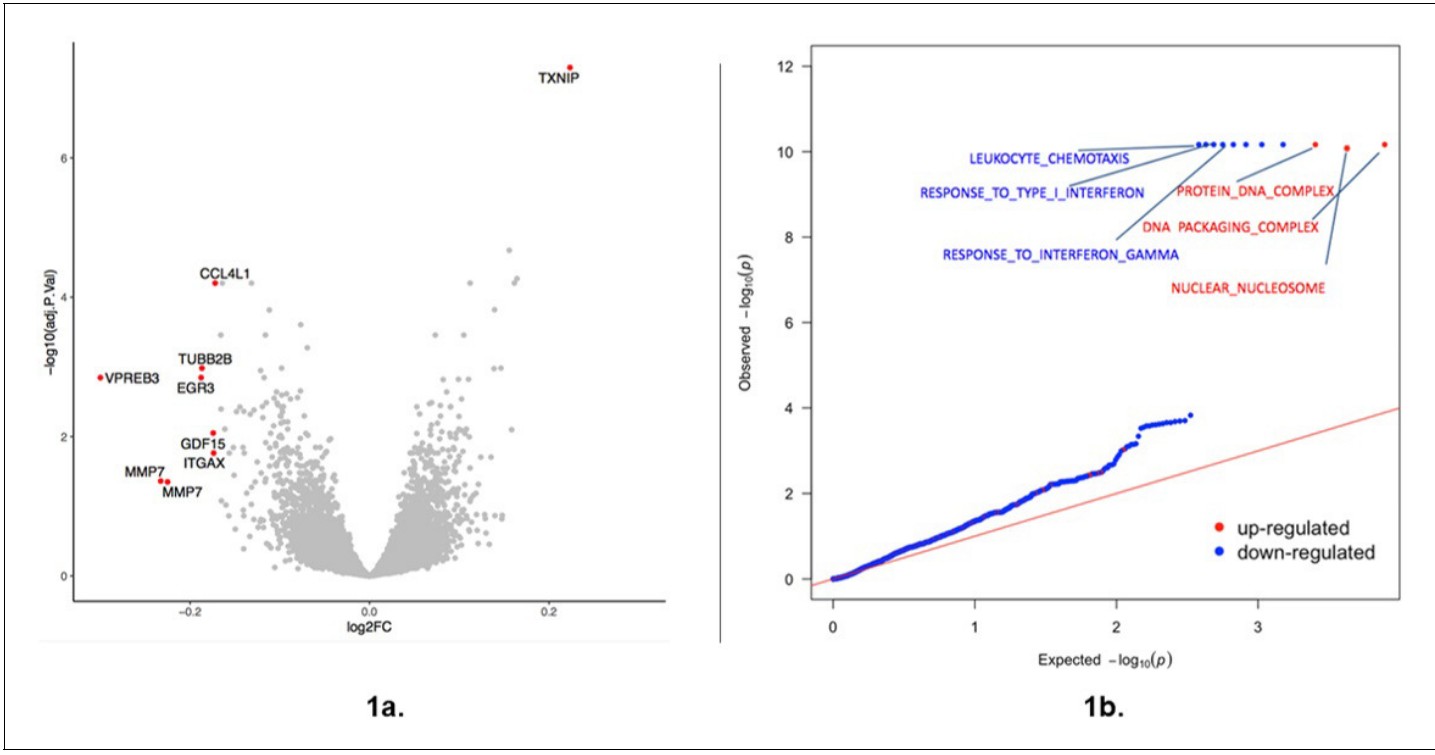

**Figure 1.** Response to glucose. (a) Volcano plot summarizing transcriptional response to glucose for all 22 individuals (RG_All consisting of nDM, nDR, and PDR individuals). Each point represents a single gene. Red indicates genes showing a differential response (FDR < 0.05; $\log_{10}$ >1.3 represented by the dotted line) and an absolute $\log_2$FC >0.17. Adj p-value is false discovery rate (FDR). FC indicates expression fold change with positive values indicating higher expression in the high glucose condition relative to the standard condition. Source and code files for this plot are available in *Figure 1—source data 1* and *Source code 1*. An additional source file can be found in Gene Expression Omnibus (GEO) at https://www.ncbi.nlm.nih.gov/geo/ under accession code GSE146615. (b) QQ (quantile-quantile) plot plot summarizing GSEA of transcriptional response to glucose in all 22 individuals. Pathways are classified as upregulated (red) or downregulated (blue) in response to glucose. Only significant GO categories (FDR < 0.1%) are labeled. Red line indicates the null expectation. Source and code files for this plot are available in *Figure 1—source data 2*, *3*, *4,* and *5* and *Source code 2*.

The online version of this article includes the following source data and figure supplement(s) for figure 1:

**Source data 1.** RG volcano plot data.
**Source data 2.** GSEA QQ plot data.
**Source data 3.** GSEA QQ plot data.
**Source data 4.** GSEA QQ plot data.
**Source data 5.** GSEA QQ plot data.
**Source data 6.** p-value distribution plot data.
**Source data 7.** Intraindividual variation HG plot.
**Source data 8.** Intraindividual variation HG plot.
**Figure supplement 1.** p-value distribution for transcriptional response to glucose in all 22 individuals (RG_All) (no diabetes, nDR, and PDR).
**Figure supplement 2.** Intra- and interindividual transcriptome variation in high glucose treatment.
**Figure supplement 3.** QQ plot demonstrating a significant shift away from the null represented by the red line of no difference between individuals as determined by assessing intra-individual variance from the biologic replicate samples.
**Figure supplement 4.** Plot showing distribution of inter vs intra-individual variance.

HG (*Shanmugam et al., 2003*). Additionally, the expression of *IL1B* is upregulated in the diabetic retina and has been implicated in the pathogenesis of diabetic retinopathy (*Liu et al., 2012*). Likewise, the top GSEA pathway has also previously been implicated in the pathogenesis of diabetic retinopathy. We identified PDGF signaling as the most significant differential response pathway (FDR = 0.012) (*Figure 2—figure supplement 2*). Elevated levels of PDGF are present in the vitreous of individuals with proliferative diabetic retinopathy (PDR) compared to individuals without diabetes (*Freyberger et al., 2000*). As PDGF is required for normal blood vessel maintenance, it is thought to contribute to the pericyte loss, microaneurysms, and acellular capillaries that are key features of

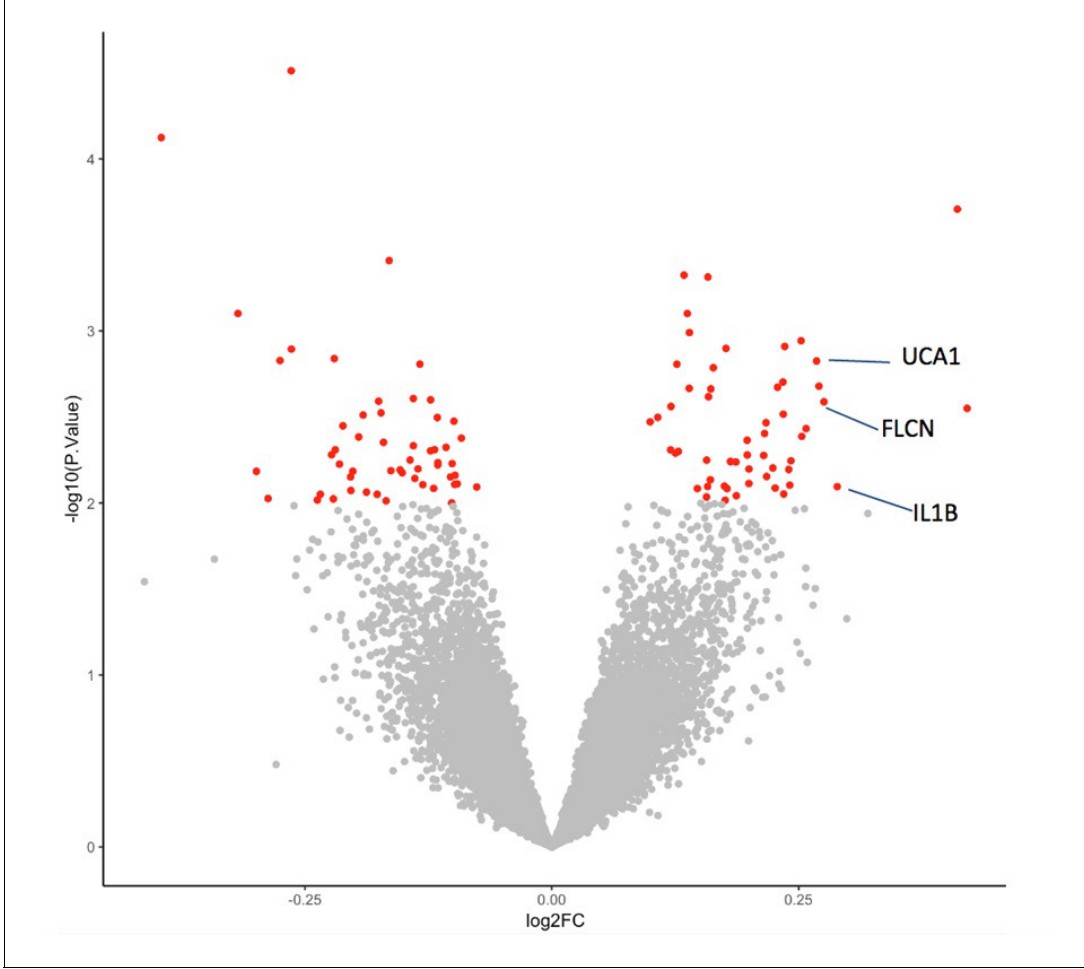

**Figure 2.** Differential transcriptional response to glucose among individuals with diabetes with and without retinopathy. Volcano plot summarizing genes exhibiting a differential response to glucose between individuals with diabetes with and without retinopathy ($RG_{PDR-nDR}$). The difference in FC between groups is represented on the X-axis and p-value of this difference on the Y-axis. Red indicates the 103 genes showing the most differential expression between individuals with and without retinopathy (p<0.01). FC, fold change. Source and code files for this plot are available in *Figure 2—source data 1* and *Source code 5*. An additional source file for this plot can be found in Gene Expression Omnibus (GEO) at https://www.ncbi.nlm.nih.gov/geo/ under accession code GSE146615.

The online version of this article includes the following source data and figure supplement(s) for figure 2:

**Source data 1.** Differential transcriptional response volcano plot.

**Source data 2.** Multidimensional scaling plot.

**Source data 3.** Multidimensional scaling plot.

**Source data 4.** Multidimensional scaling plot.

**Source data 5.** Multidimensional scaling plot.

**Figure supplement 1.** Multidimensional scaling based on differential response to glucose (rg).

**Figure supplement 2.** Gene set enrichment analysis (GSEA) of genes with differential response to glucose between individuals with diabetes with and without diabetic retinopathy.

the diabetic retina (*Hammes et al., 2002*). Interestingly, despite our model utilizing lymphoblastoid cells, it was able to reveal the upregulation of PDGF which is primarily a vascular factor that also plays a key role in neuronal tissue.

## Genetic association reveals that some genes with differential response to glucose play a role in susceptibility to diabetic retinopathy

We sought to assess whether the most significant differential response genes ($RG_{pdr-ndr}$) could yield novel insights into diabetic retinopathy. An overview of our approach is presented in *Figure 3a*.

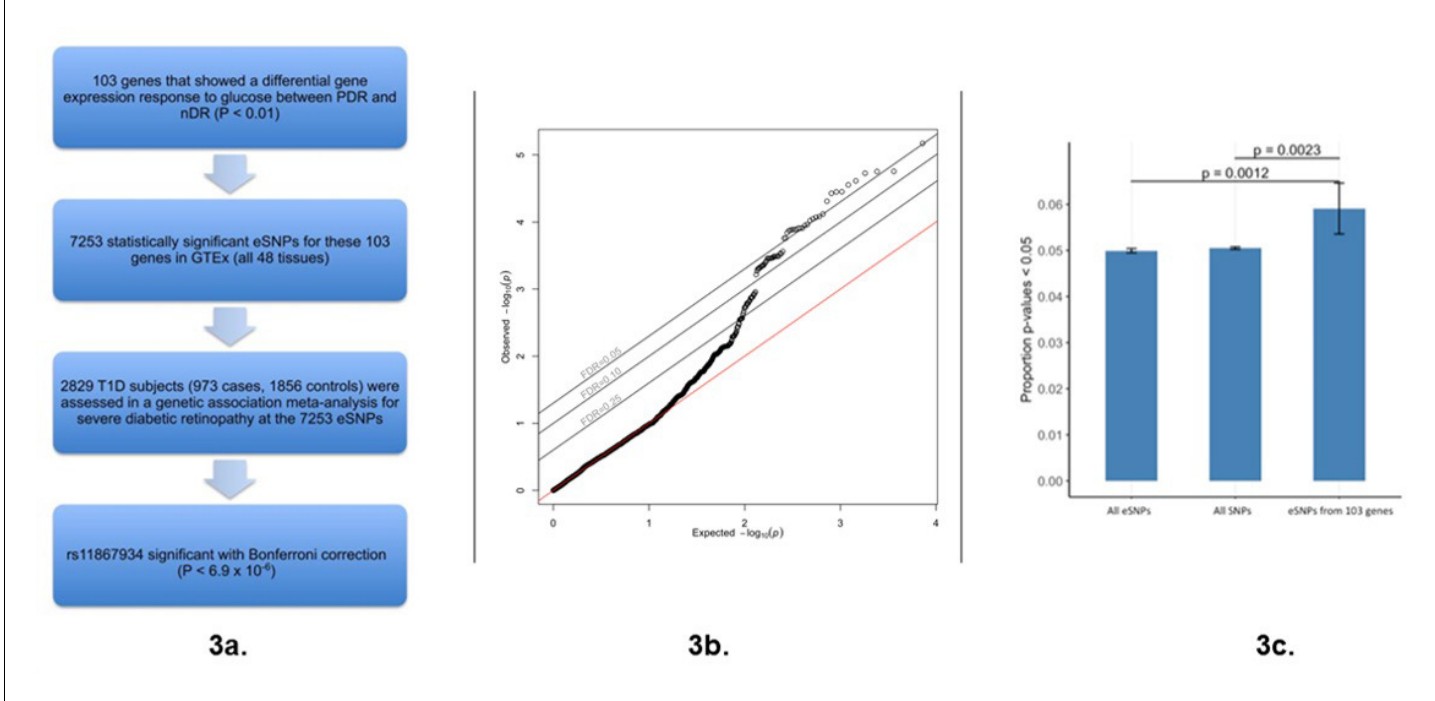

**Figure 3.** Association of glucose differential response genes (RG$_{pdr-ndr}$) with diabetic retinopathy. (a) Workflow of analytical steps integrating glucose differential response genes with genetic association with diabetic retinopathy. Flow chart showing key experimental steps based on stepwise findings. (b) QQ plot revealing a skew away from the null and above the FDR 0.05 threshold suggests that expression of some of the glucose response genes may be causally related to diabetic retinopathy. 7253 GTEx eSNPs were generated from the 103 differential response genes and tested for their association with diabetic retinopathy in a GWAS. Observed vs. expected p-values are plotted. The null hypothesis of no difference between the observed and expected p-values is represented by the red line. No influence of population structure or other design factors was observed (genomic control inflation estimate λ$_{GC}$ = 1.005) (**Devlin and Roeder, 1999**). Source and code files for this plot are available in **Figure 3—source data 1** and **Source code 7**. (c) Bar plot comparing frequency of p-values <0.05 in diabetic retinopathy GWAS of: all eSNPs, all SNPs, and eSNPs from the 103 differential response genes. An excess of GWAS p-values of <0.05 is observed in the eSNPs from the glucose differential response genes (p=0.0012 vs. all eSNPs and p=0.0023 vs. all SNPs). The proportion of SNPs with p<0.05 in the all SNPs, all eSNPs, and 103 differential response gene eSNPs are 0.0505, 0.0499, and 0.0571, respectively. Source and code files for this plot are available in **Figure 3—source data 2** and **Source code 8**.
The online version of this article includes the following source data and figure supplement(s) for figure 3:

**Source data 1.** Association of glucose differential response genes QQ plot.
**Source data 2.** Association of glucose differential response genes Bar plot.
**Source data 3.** eGenes Histogram.
**Figure supplement 1.** Enrichment of eGenes in glucose differential response genes.
**Figure supplement 2.** Histogram of frequency in which permutations of eSNPs generated from random sets of 103 genes revealed similar p-values to those generated from the set of 103 differential response genes to glucose (red dot).

First, we selected the top 103 genes (p<0.01) that showed the largest difference in gene expression response to glucose between individuals with diabetes with and without retinopathy. We next identified all of the significant expression quantitative trait loci (eQTLs) for these genes in GTEx (version 7) (**GTEx Consortium, 2015**). In total, we found 7253 unique eQTL SNPs (hereafter referred to as eSNPs) in at least one of the 48 tissues investigated by GTEx. Differential response genes are more likely to harbor eSNPs, and hence be eGenes, compared to the genome-wide average (p=2.0×10$^{-16}$) (**Figure 3—figure supplement 1**). This suggests that differential response genes are more likely to be genetically regulated and may contribute to interindividual differences in the development of diabetic retinopathy. To test if the eSNPs for the 103 differential response genes were more associated with diabetic retinopathy than expected, we evaluated the association between the 7253 differential response gene eSNPs and diabetic retinopathy using our published GWAS of diabetic retinopathy (**Grassi et al., 2011**). The 7253 eSNPs from the differential response genes are enriched for association with diabetic retinopathy (FDR < 0.05) (**Figure 3b**). To further assess the significance of this enrichment, we performed permutation testing of eSNPs from random sets of 103

genes which demonstrated that less than 1% contained the same proportion of similarly skewed GWAS p-values (*Figure 3—figure supplement 2*). The eSNPs for differential response genes were enriched among diabetic retinopathy meta-GWAS p-values relative to all eSNPs (p=0.0012) and all SNPs (p=0.0023) (*Figure 3c*). Thus, some of the genes exhibiting a differential response to glucose (RG$_{pdr-ndr}$) are associated with the development of severe diabetic retinopathy.

### *FLCN* is a putative diabetic retinopathy disease gene

The most significant retinopathy-associated eSNP among the set of 7253 eSNPs tested is rs11867934 (*Figure 4a*); FDR < 0.05; meta-GWAS p=$6.7 \times 10^{-6}$<Bonferroni adjusted p-value of $6.9 \times 10^{-6}$; OR = 0.86, 95% confidence interval (CI) = 0.71,1.00; minor allele frequency = 0.22. rs11867934 is an intergenic eSNP for *FLCN* in multiple biologically relevant tissues including artery and nerve. We confirmed FLCN expression in the retina of human donor eyes (*Figure 4—figure supplement*

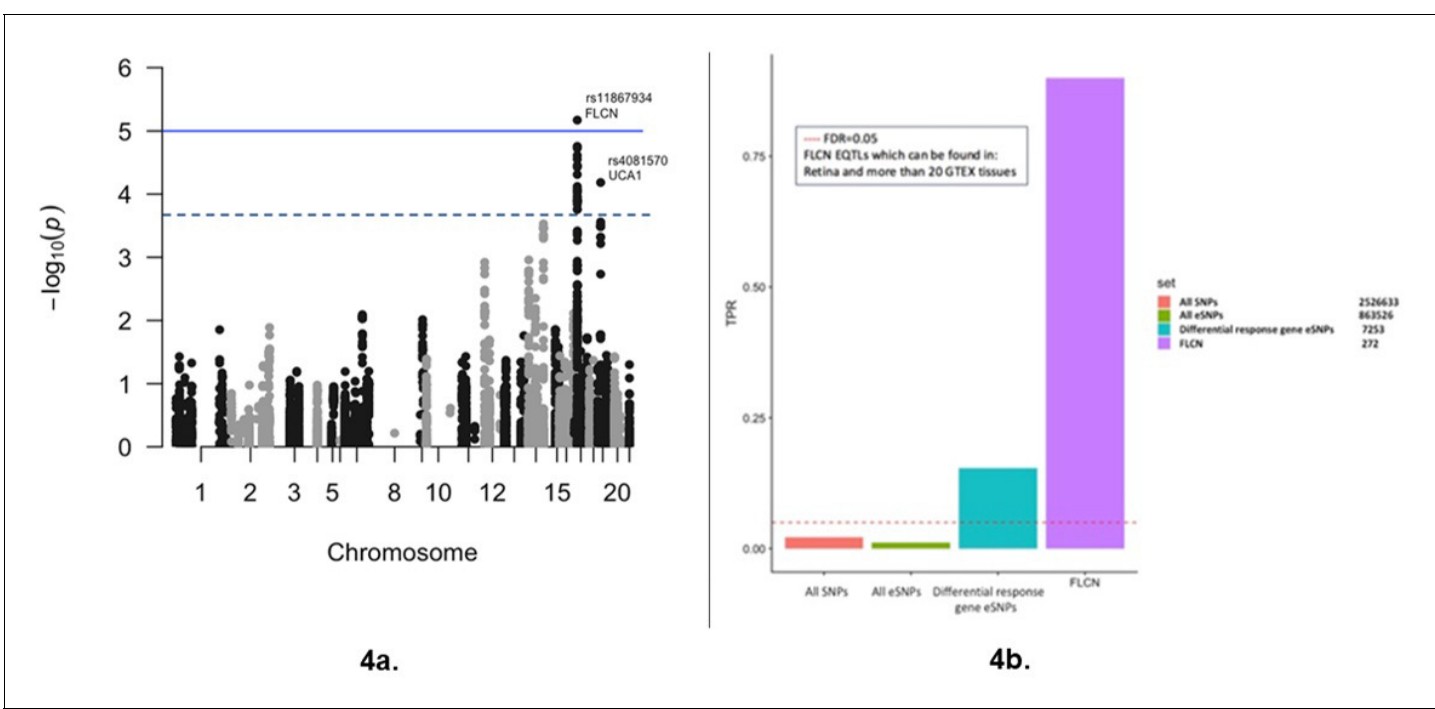

**Figure 4.** Diabetic retinopathy meta-GWAS for eSNPs of differential response genes to glucose. (a) Manhattan plot of the results of the meta-GWAS for diabetic retinopathy showing association signals for the eSNPs from the differential response genes to glucose for individuals with and without retinopathy (RG$_{PDR-nDR}$). Threshold lines represent Bonferroni correction (blue) and FDR < 0.05 (black). Association testing for diabetic retinopathy performed with 7253 eSNPs representing 103 differential response genes to glucose. Source and code files for this plot are available in *Figure 4—source data 1* and *Source code 11*. (b) Bar plot comparing the true positive rate ($\pi_1$), TPR, for association of diabetic retinopathy with all SNPs, all eSNPs, eSNPs from the 103 differential response genes to glucose (n = 7253), and eSNPs found in retina and >20 GTEx tissues for folliculin (*FLCN*) (n = 272). TPR is an estimate of the proportion of tests that are true under the alternative hypothesis. Plot reveals significant enrichment for glucose response gene eSNPs in general and for *FLCN* eSNPs ($\pi_1$ = 0.9) in particular. Source and code files for this plot are available in *Figure 4—source data 2, 3*, and *4*, and *Source code 12*.

The online version of this article includes the following source data and figure supplement(s) for figure 4:

**Source data 1.** Diabetic retinopathy meta-GWAS for eSNPs Manhattan plot.
**Source data 2.** Diabetic retinopathy meta-GWAS for eSNPs TPR plot.
**Source data 3.** Diabetic retinopathy meta-GWAS for eSNPs TPR plot.
**Source data 4.** Diabetic retinopathy meta-GWAS for eSNPs TPR plot.
**Source data 5.** Diabetic retinopathy meta-GWAS QQ plot.
**Source data 6.** Diabetic retinopathy UKBB QQ plot.
**Figure supplement 1.** Folliculin (FLCN) expression in the human retina.
**Figure supplement 2.** Folliculin (*FLCN*) expression response to glucose by disease status (PDR vs. nDR).
**Figure supplement 3.** QQ plot of diabetic retinopathy meta-GWAS p-values corresponding to 272 folliculin (FLCN) eSNPs.
**Figure supplement 4.** QQ plot of UKBB diabetic retinopathy GWAS p-values corresponding to 272 folliculin (FLCN) eSNPs.

1). In the LCLs derived from individuals with diabetes, *FLCN* was upregulated in response to glucose to a greater extent in individuals with diabetic retinopathy than in individuals with diabetes without retinopathy ($\log_2$FC difference = 0.27, p=$2.5{\times}10^{-3}$) (*Figure 2*, *Supplementary file 1b*, and *Figure 4—figure supplement 2*). eQTLs in retina have recently been mapped (*Ratnapriya et al., 2019*). We determined that at least 43% of retina eQTLs are also eQTLs in GTEx LCLs. Examining the genome-wide association signal for a disease from eQTLs in aggregate can be a more powerful strategy to discern a heterogenous genetic signal than testing each of these SNPs individually. We collated all the eSNPs for *FLCN* in the retina. We assessed the aggregated association of *FLCN* eSNPs (n = 272 eSNPs significant in the retina and 20 or more GTEx tissues) with diabetic retinopathy in the meta-GWAS and observed an enrichment for association with diabetic retinopathy ($\pi_1$ = 0.9; *Figure 4b*, *Figure 4—figure supplement 3*). We then validated the *FLCN* association with diabetic retinopathy in a third cohort, the UK Biobank (UKBB) (*Supplementary file 1c*), and found that the *FLCN* eSNPs were enriched for association with diabetic retinopathy in the UKBB ($\pi_1$ = 0.73) (*Figure 4—figure supplement 4*).

We applied Mendelian randomization to assess whether the level of *FLCN* expression affects the development of diabetic retinopathy. We first imputed retinal *FLCN* expression in the UKBB, and then estimated the effects of the estimated *FLCN* expression on diabetic retinopathy using summary data-based Mendelian randomization analysis (*Zhu et al., 2016*) (SMR). Mendelian randomization treats the genotype as an instrumental variable. A one standard deviation (SD) increase in the predicted retinal expression of *FLCN* increases the risk of diabetic retinopathy by 0.15 SD (95% CI: 0.02–0.29, standard error 0.07, p=0.024). Individuals with diabetes with high predicted retinal *FLCN* expression have increased odds of developing retinopathy (1.3 OR increase per SD increase in *FLCN* expression) (*Chinn, 2000*). We did not observe any evidence of horizontal pleiotropy (in which *FLCN* eSNPs are independently associated with both *FLCN* expression and diabetic retinopathy) confounding the analysis [HEIDI p>0.05 (p=0.2)] (*Zhu et al., 2016*). We detected an aggregated effect of 14 independent *FLCN* eQTLs ($r^2$ < 0.2) on the development of diabetic retinopathy through *FLCN* expression using multi-SNP Mendelian randomization (p=0.04) (*Wu et al., 2018a*). Together, these findings support the presence of genetic variation at the *FLCN* locus affecting both *FLCN* expression and the development of diabetic retinopathy through the expression of *FLCN*.

## Discussion

The cellular response to elevated glucose is an increasingly important pathway to understand in light of the emerging epidemic levels of diabetes worldwide (*National Diabetes Fact Sheet, 2011*). Variations in the cellular response to glucose at a molecular level have not been well characterized between cell types, and to an even lesser degree between individuals. In prior work, we characterized robust, repeatable interindividual differences in transcriptional response to glucose in LCLs of individuals with diabetic retinopathy (*Grassi et al., 2016*). As an LCL generated from each individual is genetically unique, it follows that the gene expression response to glucose between individuals should be phenotypically heterogeneous and that a portion of the interindividual variability will be genetically determined. We hypothesized that interindividual variation in the cellular response to glucose may reveal clues to the genetic basis of diabetic retinopathy, thereby providing insights into its predisposition.

We demonstrated that different individual-derived cell lines treated under identical culture conditions reveal an individual-specific transcriptional response to glucose and this signal far exceeds accompanying experimental noise. Transformation and multiple freeze/thaw passages do not homogenize the individualized response to HG-induced gene expression in LCLs. Analyzing the individual glucose-stimulated transcriptional response revealed several insights into the pathophysiology of the diabetic state and how it relates to the development of retinopathy. For instance, TXNIP was identified as the top differential response gene to glucose in all individuals (RG_all). TXNIP is a key marker of oxidative stress. It is upregulated in the diabetic retina where it induces Muller cell activation (*Devi et al., 2017*). HG treatment has been shown to increase *TXNIP* expression (*Chen et al., 2016*). *TXNIP* is a glucose sensor whose expression has been strongly associated with both hyperglycemia and diabetic complications. Specifically, the *TXNIP* locus was differentially methylated in the primary leukocytes of EDIC cases and controls (*Chen et al., 2016*). A key mechanism by which cells respond to stress is through changes in genome configuration. Conformational alterations in DNA

packaging influence the accessibility of DNA for transcription. Structural changes in DNA conformation facilitate cellular adaptation and response to stimuli which can enable transcriptional changes. The GSEA showed that the cellular response to chronic glucose stress involves alterations in DNA accessibility which facilitates the gene expression response to this environmental stimulus (*Smith and Workman, 2012*). The transcriptional response to glucose in part manifests as diminished immune responsiveness, a well-characterized feature of diabetes (*Shanmugam et al., 2003*; *Mowat and Baum, 1971*).

Further, we considered that the genetic component of an individual's response to glucose may influence their susceptibility to diabetic complications like retinopathy. Cell lines from individuals with diabetes with and without retinopathy reveal differences in the response to glucose at a molecular level. In addition, not only were some of these differential response genes biologically relevant to diabetic retinopathy as exemplified by *IL1B* and *PDGF*, but also many had a genetic basis for their differential response. By integrating the gene expression findings with GWAS data, we implicated *FLCN* as a putative disease gene in diabetic retinopathy. Mendelian randomization provided evidence that genetic variation affects diabetic retinopathy through alterations in *FLCN* expression thereby suggesting that FLCN expression is a mediator of diabetic retinopathy. *FLCN* is a biologically plausible diabetic retinopathy disease gene since its expression is present in both neuronal and vascular cells of the retina. Current evidence suggests that FLCN is a negative regulator of AMPK which helps to modulate the energy sensing ability of AMPK and plays a role in responding to cellular stress (*Hasumi et al., 2012*). AMPK plays an important role in providing resistance to cellular stresses by regulating autophagy and cellular bioenergetics to avoid apoptosis. Loss of *FLCN* results in constitutive activation of AMPK. Higher levels of FLCN would suggest less cellular capacity to deal with stress (*Possik et al., 2014*). Interestingly, the protective effect of agents such as metformin and fenofibrate on diabetic retinopathy might be mediated through AMPK (*Kim et al., 2007*; *Joe et al., 2015*).

Our study design had several advantages over prior approaches aimed at revealing the genetic basis of diabetic retinopathy. First, we utilized white blood cells which are readily accessible from the peripheral circulation of human patients (*Epidemiology of Diabetes Interventions and Complications (EDIC) Research Group, 1999*) and can reveal differential molecular characteristics depending on the stage of diabetic retinopathy (*Tang and Kern, 2011*; *Gubitosi-Klug et al., 2008*; *Kern, 2007*). LCLs are derived from white blood cells making them a relevant cellular population to study for diabetic retinopathy. LCLs have been shown to be a powerful model system for functional genetic studies in humans (*Tang and Kern, 2011*; *Kern, 2007*). Second, an LCL was generated for every individual enrolled in the landmark DCCT/EDIC study. DCCT/EDIC is the best-characterized prospective interventional cohort ever created to follow systemic complications of long-standing diabetes. DCCT/EDIC allows for detailed stratification of individuals, each of whom has had extensive prospective clinical phenotyping. Third, glucose was employed to elicit a provocative response in LCLs. By focusing on a secondary sequela of diabetes like retinopathy, the cellular response to glucose stimulation through transcription became a meaningful and directly relevant reflection of the stress each cell in the body encounters from diabetes. Insights into glucose-stimulated gene expression in LCLs have broad applicability to multiple tissues of interest for diabetic complications (even in the retina as we have shown) due to significant evidence supporting a shared framework for gene regulation among tissues (*GTEx Consortium, 2015*). Finally, disease-associated eQTL provide functional insights into the pathogenesis of a condition. We show that altering the levels of *FLCN* expression impacts risk of diabetic retinopathy. Aggregating independent eQTLs for the same gene (that are not in high linkage disequilibrium) revealed an enriched association that may otherwise have been missed by a conventional GWAS approach (*Wu et al., 2018b*). Treating the associated eQTL as an instrumental variable, Mendelian randomization supported the potential causality of *FLCN* in the pathogenesis of the disease. Inherently, this approach yielded all three M's of target modulation: mechanism, magnitude, and markers (*Plenge, 2019*).

The present work had inherent limitations. First, LCLs are not primary cells but rather a transformed cell line. The Multiple Tissue Human Expression Resource (MuTHER) LCL study revealed a large impact of common environmental exposure, stemming from shared sample handling, on gene expression in twin LCLs (*Wright et al., 2014*). The significant correlation of these extrinsic factors on LCL gene expression emphasizes the importance of randomization and biological replicates which we implemented in this study. Moreover, as a cell line, heterogeneous genomic alterations have

been identified in lymphoblastoid cells that increase with passaging, thereby raising the concern that this can lead to variability in their transcriptome (*Ben-David et al., 2018*). Importantly, the EDIC cell lines employed in this study were only passaged once previously. Additionally, genomic changes have only a minor effect on genotypic frequencies with a 99.63% genotype concordance between lymphoblastoid cells and their parent leukocytes. Mendelian error rates in levels of heterozygosity are not significantly different between LCLs and their primary B-lymphocyte cells of origin (*McCarthy et al., 2016*). Second, it is not possible to delineate cause from effect in gene expression studies. Gene expression changes may be causal, epiphenomena, or due to reverse causality (the disease causing the gene expression changes rather than the other way around). In this study, by integrating genetic analyses with gene expression and recognizing that variation in the underlying genome precedes disease onset and can therefore be considered an instrumental variable, we identified through Mendelian randomization potentially causal gene expression changes in *FLCN* that act as a mediator for retinopathy thereby avoiding the trap of reverse causality. Finally, eQTL found in LCLs may not be relevant to diabetic retinopathy. As noted previously we found 43% of retina eQTL are shared with LCLs. We demonstrated that independent *FLCN* eQTLs found both in the retina and GTEx tissues showed an enriched association with diabetic retinopathy, a finding that was replicated in a large independent cohort from the UKBB. For complex trait associations in general and for those specifically in the retina, eQTL that are shared between tissues explain a greater proportion of associations than tissue-specific eQTL (*Gamazon et al., 2018*). For instance, shared tissue eQTL are enriched among genetic associations with age-related macular degeneration, another common retinal disease, despite the high tissue specificity of the disease (*Ratnapriya et al., 2019*; *Unlu et al., 2019*).

In summary, integration of gene expression from a relevant cellular model with genetic association data provided insights into the functional relevance of genetic risk for a complex disease. Using disease-associated differential gene and eQTL-based genome-wide association testing, we identified possible causal genetic pathways for diabetic retinopathy. Specifically, our studies implicated *FLCN* as a putative diabetic retinopathy susceptibility gene. Future work that incorporates more extensive molecular profiling of the cellular response to glucose in conjunction with a greater number of cell lines may yield further insights into the underlying genetic basis of diabetic retinopathy.

## Materials and methods

### Overview

In this study we profiled the transcriptomes of cell lines derived from 22 individuals (seven individuals with no diabetes [nDM], eight with T1D with PDR, and seven with T1D with no retinopathy [nDR]) utilizing gene expression microarrays to characterize the transcriptional response to glucose. Specifically, we cultured LCLs derived from each individual in SG and HG medium and measured gene expression for each gene in each sample, as well as the difference ($\Delta$ = response to glucose [RG]) in each gene's expression for each individual (*Figure 5a*). We compared the differential response in gene expression to glucose for all individuals with and without proliferative retinopathy. '*Differential response*' in gene expression refers to the difference in gene expression response to glucose between groups. Specifically, we identified genes with a significant differential response in expression between individuals with diabetes with and without PDR ($RG_{pdr-ndr}$). We followed up genes showing differential response using the results of both a prior genome-wide association study (GWAS) meta-analysis of diabetic retinopathy (in the GoKinD and EDIC cohorts) (*Grassi et al., 2011*) and the results of a multi-tissue eQTL analysis from GTEx (*GTEx Consortium, 2015*) to identify potential diabetic retinopathy susceptibility genes (*Figure 5b*).

### Participant safety and confidentiality issues

All cell lines were de-identified prior to their arrival at the University of Illinois at Chicago; therefore, this proposal qualified as non-human subjects research according to the guidelines set forth by the institutional review board at the University of Illinois at Chicago. As the data were analyzed anonymously, no participant consent was required. DCCT participants previously provided consent for their samples to be used for research. Matching of participants was performed at George Washington University Biostatistics Center and did not involve protected health information as the

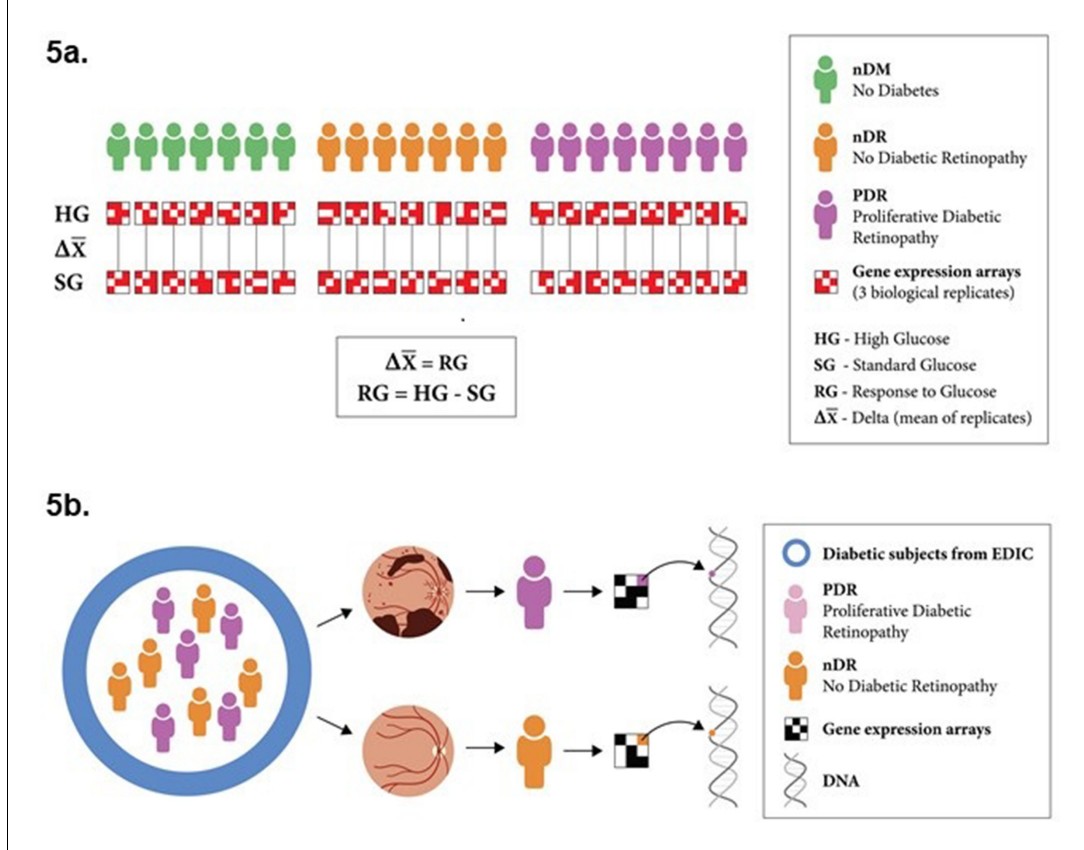

**Figure 5.** Experimental design. (a) Schematic representation of the experimental design for transcriptomic profiling. Lymphoblastoid cell lines (LCLs) from 22 individuals were cultured under both standard glucose (SG) and high glucose (HG) conditions. Gene expression was quantified using microarrays for three biological replicates of each LCL in each condition. The response to glucose was determined for all genes on a per-individual basis, by comparing expression in SG and HG conditions. The cell lines were derived from individuals with diabetes and no retinopathy (7), individuals with diabetes and proliferative diabetic retinopathy (8), and individuals without diabetes (7). (b) We identified 15 individuals based on retinopathy status from the Epidemiology of Diabetes Interventions and Complications (EDIC) cohort. We compared the differential response in gene expression to glucose for individuals with and without proliferative retinopathy ($RG_{pdr-ndr}$). Expression quantitative trait loci (eQTL) for those genes that showed the greatest differential response between individuals with and without retinopathy were tested for their genetic association with diabetic retinopathy.

phenotypic data were de-identified. The George Washington University institutional review board has approved all analyses of EDIC data of this nature. All protocols used for this portion of the study are in accordance with federal regulations and the principles expressed in the Declaration of Helsinki. Specific approval of the study design and plan was obtained from the EDIC Research Review Committee.

## Cell lines

Twenty-two LCLs were used in the study as described previously (*Grassi et al., 2016*). Briefly, we included 15 of the 1441 LCLs generated from individuals with type 1 diabetes from the DCCT/EDIC cohort (*The Writing Team for the Diabetes Control and Complications Trial/Epidemiology of Diabetes Interventions and Complications Research Group, 2002*; *Epidemiology of Diabetes Interventions and Complications (EDIC) Research Group, 1999*), consisting of eight matched cases with PDR and seven without retinopathy (nDR) as the controls (*Grassi et al., 2016*; *Supplementary file 1d*). Whole blood samples were ascertained from DCCT study participants between 1991 and 1993. White blood cells from the samples were transformed into LCLs in the early 2000s. The 15 LCLs from individuals with diabetes consisted of matched cases and controls. Cases were defined by the development of PDR by EDIC Year 10 (2004), whereas controls had no retinopathy through EDIC Year 10 (2004). Retinopathy grade was determined by seven-field stereoscopic photos. Control participants had an ETDRS (Early Treatment Diabetes Retinopathy Score) of 10 and case participants had an

ETDRS score of $\geq$61. All eight pairs were matched by age, sex, treatment group (intensive vs. conventional), cohort (primary vs. secondary), and diabetes duration (*The DCCT Research Group, 1986*; *The diabetes control and complications trial, 1995*), except one pair that was matched by age, sex, and treatment group only. Diabetes duration was defined as the number of months since the diagnosis of diabetes at DCCT baseline which was the time at participant enrollment (1983–1989). For the seven pairs matched on duration, four pairs were matched by duration quartiles (baseline duration 0–4 years, 4–8 years, 8–12 years, or 12–15 years) and three pairs were matched by duration halves (<8 years vs. $\geq$ 8 years). Matching by age was done similar to duration: four pairs by quartile (<21 years, 21–25 years, 26–31 years, and $\geq$31 years) and the remaining four by halves (<26 years vs. $\geq$26 years). The remaining seven LCLs were purchased from the Coriell Institute for Medical Research NIGMS Human Genetic Cell Repository (http://ccr.coriell.org/) (GM14581, GM14569, GM14381, GM07012, GM14520, GM11985, and GM07344). None of these individuals had a history of diabetes (nDM). The covariates available for these seven individuals were age and sex; male and female individuals were included. All of these individuals without diabetes were unrelated and of European ancestry (*Grassi et al., 2016*; *Grassi et al., 2014*; *Supplementary file 1d*). All 22 cell lines underwent Hoechst staining to ensure they were free from mycoplasma contamination.

## Culture conditions

All LCLs were maintained in conventional lymphocyte cell culture conditions of RPMI 1640 with 10% FBS in a 25 cm$^2$ cell culture flask. The cells were incubated at 37°C in 5% $CO_2$ and the media was changed twice each week. Prior to the experiments (below), lymphoblastoid cells following serum starvation were passaged for a minimum of 1 week in either SG RPMI 1640 (11 mM glucose) or HG RPMI media (30 mM glucose) (*Caramori et al., 2012*).

## Gene expression profiling

Quality control from RNA extraction was performed using the Agilent bio-analyzer, processed using the Illumina TotalPrep$-$96 RNA Amplification Kit (ThermoFisher 4393543), hybridized to Illumina HT12v4 microarrays (Catalog number: 4393543), and scanned on an Illumina HiScan scanner (*Du et al., 2008*; *Lin et al., 2008*). For each of the 22 individuals, three biological replicates were profiled, with each sample assessed at both SG conditions (11 mM of glucose) and HG conditions (30 mM of glucose). Biological replicates were split from the same mother flask; cells were grown in separate flasks and run on different microarray plates on different days. Each biological replicate was generated from a separate frozen aliquot of that cell line. The gene expression profiling was performed in a masked fashion for both the case/control (PDR, nDR, and nDM) status of the individual and the glucose treatment (SG and HG) of the sample in order to reduce any bias.

## Relative EBV copy number

Standard TaqMan qPCR was performed using EBV and NRF1 probes and primers (*Choy et al., 2008*). To calculate real-time PCR efficiencies a standard curve of 10 points of twofold dilution of 156.7 ng of gDNA was used from the Raji cell line (ATCC CCL-86). Probes were designed for the target, EBV, and a reference gene, *NRF1*. Final concentrations of the probes and primers were 657 nM and 250 nM, respectively. EBV probe: 5'6FAM-CCACCTCCACGTGGATCACGA-MGBNFGQ3'; EBV forward primer: 5' GAGCGATCTTGGCAATCTCT; EBV reverse primer: 5' AGTAGCCAGGCACCTAC TGG; NRF1 probe: 5'VIC-CACTGCATGTGCTTCTATGGTAGCCA-MGBNFQ3'; NRF1 forward primer: 5' ATGGAGGAACACGGAGTGAC; NRF1 reverse primer: 5' CATCAGCTGCTGTGGAGTTG. Cycle number of crossing point versus DNA concentration were plotted to calculate the slope. The real-time PCR efficiency (E) was calculated according to the equation: $E = 10^{(-1/\text{slope})}$. Triplicates were done for each data point. Genomic DNA (78.3 ng) from each LCL was used in a standard TaqMan qPCR reaction with *EBV* as target gene and *NRF1* as reference gene. The sequences and concentrations of the probes and primers were as shown above.

## Growth rate measurement

LCLs were thawed and cultured in RPMI and 10% FBS until they reached over 85% cell viability. Cells were seeded in a T25 flask. Two replicates were performed per cell line. Cells were counted every day or every other day for 5–10 days and recorded.

## Quality control for gene expression

The gene expression data comprised a total of 144 samples from 22 individuals (three replicates per individual and treatment, except for three individuals with five replicates). Gene expression was assessed in two conditions, SG and HG, and generated in four batch runs that were carefully designed to minimize potential batch effects. BeadChip data were extracted using GenomeStudio (version GSGX 1.9.0) and the raw expression and control probe data from the four different batches were preprocessed using a lumiExpresso function in the lumi R package version 2.38.0 (*Grassi et al., 2011*; *Grassi et al., 2012*) in three steps: (i) background correction (lumiB function with the bgAdjust method); (ii) variance stabilizing transformation (lumiT function with the log2 option); and (iii) normalization (lumiN function with the robust spline normalization [rsn] algorithm that is a mixture of quantile and loess normalization). To remove unexpressed probes, we applied a detection filter to retain probes with strong true signal by applying Illumina BeadArrays detection p-values <0.01 followed by removing probes that did not have annotated genes, resulting in a total of 15,591 probes.

## Gene expression analysis

The study design is portrayed in *Figure 5a*. For a given individual $S_i$ ($i$ = 1,...,22) and gene $G_k$ ($k$ = 1,...,15591), we calculated $\Delta_{i,k}$ = $HG_{i,k}$− $SG_{i,k}$, where $\Delta$ is the individual's response to glucose (RG), HG is gene expression in high glucose culture, and SG is gene expression in standard glucose culture. All replicate data were fit using a mixed model that accounted for the correlation between repeated measures within individuals. The design matrix was constructed and analysis performed using the R version 3.5.1 package *limma* (*Ritchie et al., 2015*). We built a design matrix using the model.matrix function, and accounted for correlation between biological triplicates using limma's *duplicate correlation* function. A mixed linear model was then fit that incorporates this correlation and $\Delta_{i,k}$ using the *lmFit* function. PCA of gene expression was run with the *prcomp* function in R (*Becker, 1988*). For each gene, we calculated moderated t- and f-statistics and log-odds of expression by empirical Bayes moderation of the standard errors toward a common value. Differential response reflects fold change (FC) differences between matched individuals in the two groups in their paired response to glucose. The power to detect a 2 FC difference in gene expression between the two retinopathy groups (retinopathy vs. no retinopathy) ($RG_{pdr−ndr}$) with a paired analysis given our sample size and using a type I error rate of 0.05 is 95% (as supported by our prior work *Grassi et al., 2016*).

## Gene set enrichment analysis

GSEA was performed using pre-ranked gene lists (*Subramanian et al., 2005*). We ranked all analyzed genes based on sign (fold change) $\times$ (–$\log_{10}$(p-value)) (*Grassi et al., 2012*). Duplicated genes were removed. The gene ranking resulted in the inclusion of 11,579 genes. Enrichment statistics were calculated using rank weighting and the significance of enrichment was determined using permutations performed by gene set. The gene sets included c2.all.v6.0 and c5.all.v6.0. The minimum gene set size was 15 and the maximum gene set size was 500. GSEA was used to identify significant gene sets for the response to glucose in all study participants ($RG_{all}$: nDM + PDR + nDR).

## Expression quantitative trait loci

To determine if the genes showing a differential response in gene expression ($RG_{pdr−ndr}$) is driven by germline genetic variation, we tested if the eQTLs for these genes are enriched for small diabetic retinopathy GWAS p-values (*Grassi et al., 2011*). We use the term 'differential response gene' for those genes identified by the $RG_{pdr−ndr}$ analysis. All statistically significant eSNPs (false discovery rate [FDR] threshold of ≤0.05) (single nucleotide polymorphisms, SNPs, corresponding to *cis*-eQTLs from the GTEx and EyeGEx datasets) were collated for the glucose response genes in any of the 48 GTEx (version 7) tissues and the retina (*GTEx Consortium, 2015*; *Ratnapriya et al., 2019*). We use the term eGene for any gene with at least one significant eSNP in any tissue.

## Genome-wide association study

Meta-analysis p-values were ascertained from our prior GWAS for diabetic retinopathy (*Grassi et al., 2011*). The study assessed the genetic risk of sight threatening complications of diabetic retinopathy as defined by the presence of diabetic macular edema or PDR (cases) compared to those without

(controls) in two large type 1 diabetes cohorts of 2829 total individuals (973 cases and 1856 controls) taken from the Genetics of Kidney in Diabetes (GoKinD) and the Epidemiology of Diabetes Interventions and Complications study (EDIC) cohorts.

We sought to determine whether there is enrichment of small p-values for diabetic retinopathy meta-GWAS among the significant eQTLs for the glucose response genes that show a significant differential glucose response between individuals with and without retinopathy (RG$_{pdr-ndr}$). We used Benjamini–Hochberg adjusted p-values (FDR) to account for multiple testing given the high level of linkage disequilibrium between many eSNPs within an eQTL. SNPs from the three studies (expression, eQTL, and GWAS) were matched by mapping all SNPs to dbSNP v.147 (*Grassi et al., 2013*). We determined the corresponding FDR for each glucose response gene's eSNPs in the diabetic retinopathy meta-GWAS. The Bonferroni correction was used to establish the threshold for significance. To assess enrichment, we first determined the observed proportion of meta-GWAS FDR values <0.05 among the statistically significant eQTLs of the glucose response genes (RG$_{pdr-ndr}$). Next, we took 10,000 random samples of 103 GTEx eGenes (genes with an eQTL in any GTEx tissue) and identified corresponding eSNPs across all GTEx tissues. We calculated the GWAS FDR for these eSNPs and recorded the proportion of FDR values <0.05.

Validation for the association of glucose response gene eSNPs with diabetic retinopathy was performed in the UKBB GWAS (*Supplementary file 1c*; *Sudlow et al., 2015*). Only individuals of northern European ancestry were analyzed. Quality control excluded individuals who were outliers based on relatedness, exhibited an excess of missing genotype calls, had more heterozygosity than expected, or had sex chromosome aneuploidy. A total of 337,147 individuals were available for analysis. Case participants were defined as those who answered 'yes' to questionnaire data eyesight field 6148 'Diabetes related eye disease' (n = 2332). Our prior work validated the utility of self-report for the presence of severe diabetic retinopathy (*Grassi et al., 2013*; *Grassi et al., 2009*). Control participants were defined as those who answered 'yes' to data field 2443 'Diabetes diagnosed by doctor' (n = 14,680), excluding case participants. SNPs were excluded according to the following: minor allele frequency <0.004; missing rate >0.015; HWE p<1×10$^{-10}$; INFO score <0.8. We performed logistic regression as implemented in Plink2 (*Chang et al., 2015*) on this set of cases and controls. The logistic regression, including the following covariates: first 10 genotype-based principal components, chromosomal sex (as defined by XX, XY status), age, type of diabetes, HbA1c, and genotyping array type.

## Mendelian randomization

To explore a possible causal effect of increased *FLCN* expression on diabetic retinopathy, we employed Mendelian randomization (*Davies et al., 2019*). Effects were estimated with summary data-based Mendelian randomization analysis (*Zhu et al., 2016*) (SMR). We estimated the effect of increasing levels of *FLCN* expression on diabetic retinopathy in the UKBB GWAS for diabetic retinopathy (described above) utilizing 272 SNPs that were significant cis-eSNPs (FDR ≤ 0.05) for *FLCN* in retina and also in at least 20 GTEx tissues. A total of 246 SNPs remained after removing those SNPs or their proxies (r$^2$ > 0.8) not genotyped in the UKBB. For each individual, the exposure was based on the genetically predicted gene expression of *FLCN* in retina and the outcome was the likelihood of having diabetic retinopathy. Heterogeneity in dependent instruments (HEIDI) (*Zhu et al., 2016*) was used to investigate the possibility of confounding bias from horizontal pleiotropy with 14 independent (r$^2$ < 0.2) *FLCN* eQTLs. As multiple independent (r$^2$ < 0.2, n = 14) *FLCN* eQTLs exist, we also employed multi-SNP Mendelian randomization to assess for an aggregated effect (*Wu et al., 2018a*) of the eQTLs on diabetic retinopathy mediated through *FLCN* expression.

## FLCN localization in human donor eye retina

A whole eye from a 69-year-old Caucasian female post-mortem donor without diabetes was obtained from National Disease Research Interchange (NDRI). Findings were replicated in an additional five post-mortem donors without diabetes from the NDRI. The eye was cut in half in a horizontal plane, and each half was placed in an individual cassette. Samples were processed on ASP300 S automated tissue processor (Leica Biosystems) using a standard overnight processing protocol and embedded into paraffin blocks. Tissue was sectioned at 5 μm, and sections were de-paraffinized and stained on BOND RX autostainer (Leica Biosystems) following a preset protocol. In brief, sections

were subjected to EDTA-based (BOND ER2 solution, pH9) antigen retrieval for 40 min at 100°C, washed, and incubated with protein block (Background Sniper, Biocare Medical, BS966) for 30 min at room temperature. For immunofluorescence (IF), sequential staining with rabbit polyclonal anti-FLCN antibody (1:50, Abcam #ab93196) and mouse monoclonal anti-CD31 antibody (1:50, DAKO, M0823) was conducted using goat-anti-rabbit Alexa-488 and goat-anti-mouse Alexa-555 secondary antibodies (Molecular Probes) for detection. DAPI (Invitrogen, #D3571) was used to stain nuclei. The slides were mounted with ProLong Diamond Antifade mounting media (ThermoFisher, #P36961). Images were taken at 20× magnification on Vectra three multispectral imaging system (Akoya Biosciences). A spectral library acquired from mono stains for each fluorophore (Alexa-488, Alexa-594), DAPI, and human retina background fluorescence slide was used to spectrally unmix images in InForm software (Akoya Biosciences) for visualization of each color.

## Acknowledgements

This research has been conducted using the UK Biobank Resource under Application Number 44316. We acknowledge the guidance and assistance provided by the members of the DCCT/EDIC committee at the time of this publication. A complete list of investigators and members of the Research Group appears in *DCCT/EDIC Research Group et al., 2017*. We thank the DNA Services Facility and the Research Histology and Tissue Imaging Core at UIC Research Resources Center for assistance in histological techniques and image acquisition. We thank Andrew D Paterson, MD, for helpful input and comments.

## Additional information

### Funding

| Funder | Grant reference number | Author |
| --- | --- | --- |
| National Eye Institute | R01EY023644 | Michael A Grassi |
| National Eye Institute | ZIAEY000546 | Anand Swaroop |

The funders had no role in study design, data collection and interpretation, or the decision to submit the work for publication.

### Author contributions

Andrew D Skol, Data curation, Software, Formal analysis, Validation, Methodology, Writing - review and editing; Segun C Jung, Data curation, Software, Formal analysis, Validation, Writing - review and editing; Ana Marija Sokovic, Data curation, Software, Formal analysis, Validation; Siquan Chen, Maria Sverdlov, Investigation, Methodology; Sarah Fazal, Investigation; Olukayode Sosina, Data curation, Formal analysis; Poulami P Borkar, Resources, Investigation, Project administration, Writing - review and editing; Amy Lin, Formal analysis; Dingcai Cao, Software, Formal analysis; Anand Swaroop, Formal analysis, Supervision, Writing - review and editing; Ionut Bebu, Resources, Software, Formal analysis; DCCT/EDIC Study group, Resources; Barbara E Stranger, Conceptualization, Data curation, Formal analysis, Supervision, Validation, Visualization, Methodology, Writing - review and editing; Michael A Grassi, Conceptualization, Resources, Formal analysis, Supervision, Funding acquisition, Investigation, Visualization, Methodology, Writing - original draft, Project administration, Writing - review and editing

### Author ORCIDs

Anand Swaroop [ID] http://orcid.org/0000-0002-1975-1141
Michael A Grassi [ID] https://orcid.org/0000-0002-8467-3223

### Decision letter and Author response

Decision letter https://doi.org/10.7554/eLife.59980.sa1
Author response https://doi.org/10.7554/eLife.59980.sa2

# Additional files

## Supplementary files

- Source code 1. RG volcano plot.
- Source code 2. GSEA QQ plot.
- Source code 3. p-value distribution plot.
- Source code 4. Intraindividual variation HG plot.
- Source code 5. Differential transcriptional response volcano plot.
- Source code 6. Multidimensional scaling plot.
- Source code 7. Association of glucose differential response genes QQ plot.
- Source code 8. Association of glucose differential response genes Bar plot.
- Source code 9. Enrichment of eGenes plot.
- Source code 10. eGenes Histogram.
- Source code 11. Diabetic retinopathy meta-GWAS for eSNPs Manhattan plot.
- Source code 12. Diabetic retinopathy meta-GWAS for eSNPs TPR plot.
- Source code 13. FLCN expression response to glucose by disease status box and whisker plot.
- Source code 14. Diabetic retinopathy meta-GWAS QQ plot.
- Source code 15. Diabetic retinopathy UKBB QQ plot.
- Source data 1. Differential response to Glucose PDR vs nDR.

- Supplementary file 1. Supplementary file 1a. Demographic features of the DCCT/EDIC study subjects with type 1 diabetes. Source files can be found at Epidemiology of Diabetes Interventions and Complications (EDIC). Design, implementation, and preliminary results of a long-term follow-up of the Diabetes Control and Complications Trial cohort. Diabetes Care. 1999 Jan;22(1):99–111. pmid:10333910. Pubmed Central PMCID: 2745938. Epub 1999/05/20. eng. Supplementary file 1b. Differential response to Glucose PDR vs. nDR ($RG_{pdr-ndr}$). Source file for this table is available at *Source data 1*. Supplementary file 1c. Demographic features of UK Biobank subjects with diabetes used in the diabetic retinopathy GWAS. Source file for this table is available at https://www.ukbiobank.ac.uk/. Supplementary file 1d. Demographic features of individuals without diabetes from the Coriell Institute for Medical Research NIGMS Human Genetic Cell Repository. Source file for this table is available at Coriell Institute for Medical Research NIGMS Human Genetic Cell Repository (http://ccr.coriell.org/).

- Transparent reporting form

## Data availability

Source files and code for all the figures and tables have been provided, except for drawings, flowcharts and histopathology findings. We have also included links and references where appropriate. Figure 3 source data 5 and 6 are available on Dryad at https://doi.org/10.5061/dryad.zkh18938j. Additional data files can be found here: microarray expression data at Gene Expression Omnibus (GEO) under accession code GSE146615 and diabetic retinopathy GWAS data at UKBB archive (https://biobank.ctsu.ox.ac.uk/crystal/docs.cgi?id=1).

The following datasets were generated:

| Author(s) | Year | Dataset title | Dataset URL | Database and Identifier |
|---|---|---|---|---|
| Skol A, Jung SC, Sokovic AM, Chen S, Fazal S, Sosina O, Borkar PP, Lin A, Sverdlov M, Cao D, Swaroop A, Bebu I, Stranger BE, Grassi MA | 2020 | Data from: Integration of genomics and transcriptomics predicts diabetic retinopathy susceptibility genes | https://doi.org/10.5061/dryad.zkh18938j | Dryad Digital Repository, 10.5061/dryad.zkh18938j |

| Sokovic AM, Grassi MA | 2020 | Mendelian randomization identifies FLCN expression as a mediator of diabetic retinopathy | https://www.ncbi.nlm.nih.gov/geo/query/acc.cgi?acc=GSE146615 | NCBI Gene Expression Omnibus, GSE146615 |
|---|---|---|---|---|

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
