## [Decision Letter]

**Acceptance summary:**

This study addresses the genetic underpinnings of diabetic retinopathy one of the major complications of diabetes in humans. The approach used here was considered innovative. Rather than using traditional genetic approaches, such as genome wide association, the examination of glucose induced changes in gene expression in cell lines from people with type 1 diabetes either with or without retinopathy provided new insights into the disease. The identification of SNPs associated with such changes – so called eQTLs – followed by validation in independent human cohorts took the study one step beyond many others in the field. The further confirmation of *FLCN* as a mediator of diabetic retinopathy using Mendelian Randomization provided further confidence in the method. This study was considered to provide advances in our understanding of the mechanisms that lead to diabetic retinopathy in humans.

**Decision letter after peer review:**

Thank you for submitting your article "Mendelian randomization identifies folliculin expression as a mediator of diabetic retinopathy." for consideration by *eLife*. Your article has been reviewed by two peer reviewers, and the evaluation has been overseen by David James as the Senior and Reviewing Editor. The reviewers have opted to remain anonymous.

The reviewers have discussed the reviews with one another and the Reviewing Editor has drafted this decision to help you prepare a revised submission.

While your study was well performed, several rather serious issues were raised by the reviewers as detailed below and all of these points need to be addressed prior to further consideration by *eLife*. Also one reviewer noted that the title only focused on "Mendelian randomization", which is an overstatement of what is essentially only a gene expression study. In addition stating that RM " identifies folliculin expression as a mediator of diabetic retinopathy" is also an overstatement as the mediator effect is not shown. So it will be important to reword the title if you decide to undertake a substantial revision.

Summary:

This study investigates gene expression profiling related to diabetic retinopathy using several strategies including differential gene expression associated with response to glucose by comparing lymphoblastoid cell lines (LCLs) between cases (with retinopathy) and controls (without retinopathy) with type 1 diabetes. The study identified significant eQTLs from gene expression analysis and public gene expression databases and then tested significant eSNPs by the meta-analysis GWAS using independent cohorts. The expression of one gene, *FLCN*, to be a mediator of diabetic retinopathy by the Mendelian Randomization method was confirmed.

Essential revisions:

1) The whole paper and its conclusions are based on a very small number of samples and not supported by strong experimental data about causality. Overall, the small group of studied subjects present huge differences in duration of diabetes and glucose control the 2 main factors of retinopathy. Thus, it is unclear how to differentiate the biological effects of long term high glucose and their impact on retinopathy.

2) Based on the transcriptome analysis the conclusion "This finding suggests that chronic glucose exposure depresses cellular immune responsiveness and may explain in part the increased risk of infection found in patients with diabetes" is not based on evidence as authors selected transcripts of their choice and also causality is not shown.

"Individuals with diabetic retinopathy exhibit a differential transcriptional response to glucose". Note that the level of association shown (especially for PDGF) is somewhat marginal.

3) "Genes with differential response to glucose are implicated in the pathogenesis of diabetic retinopathy." This part is the most intriguing and original but it is based on expression in many tissues and thus the title is also overstated: it shows some kind of association but certainly not that the 103 genes "are implicated" in retinopathy.

4) "Folliculin (*FLCN*) is a putative diabetic retinopathy disease gene" this part is also interesting (and includes some in vivo experiments) but the original whole genome gene expression study did not detect FLNC as differentially expressed in the cell blood of the patients with retinopathy. Can the authors comment on this? One of the reviewers also mentioned that no SNPs near *FLCN* have been identified in diabetes (or complications) GWAS and this is somewhat worrying.

5) It is confusing that the authors used different selection criteria for gene identifications. In Results (subsection “Individuals with diabetic retinopathy exhibit a differential transcriptional response to glucose”), they identified 19 differentially response genes (P <0.05) between retinopathy cases and controls. However, they have selected the top 103 genes with P<0.01 (Results, subsection “Genes with differential response to glucose are implicated in the pathogenesis of diabetic retinopathy”) for further investigations. What is the gap between these two gene sets selection? I assume that the *FLCN* gene is in the top 103 gene set but not in the above 19 gene set. Explanations are needed for including specific genes for different analysis purposes.

6) The authors selected LCLs from individuals of 3 groups, non-diabetes (nDM), type 1 diabetes without retinopathy (nDR) and type 1 diabetes with proliferative diabetic retinopathy (PDR). The benefits of utilizing nDM samples in the analysis was not clear. Although both gene expression and GSEA methods were conducted, the results were not relevant to diabetic retinopathy. What is the purpose of including these samples? Similarly, it is not clear what the purpose of using the gene set enrichment analysis (GSEA) was because it seems that the authors performed most analyses to identify genetic components by gene-based or SNP-based methods in the manuscript.

7) The authors tested gene expression profile and associations using data from type 1 diabetic retinopathy. However, for the confirmation with UK BioBank (UKBB) data, they included all samples with both type 1 and type 2 diabetes. Did you perform the analysis stratified by the type of diabetes? Do you have any explanations of possible differences?

---

## [Author Response]

Essential revisions:1) The whole paper and its conclusions are based on a very small number of samples and not supported by strong experimental data about causality.

The gene expression analyses conducted in our work were performed for the purposes of discovery and hypothesis generation, which were then to be subsequently validated in the genomic studies. While we agree with the reviewer’s observation that the transcriptomic studies were conducted in a small group of individuals, it is important to note that the conclusions of the manuscript are not based on the gene expression findings themselves, but rather on the genomic analyses which incorporated close to twenty thousand individuals from three independent cohorts.

Genetics offers an excellent way to differentiate the biologic effects of long-term high glucose and its impact on retinopathy thereby disentangling causality, epiphenomenon and reverse causality (Pearl, J, Cambridge University Press, 2009; Davies et al., 2019). Recognizing that variation in the underlying genome precedes disease onset and can therefore be considered an instrumental variable, we integrated genetic analyses with gene expression and identified, through Mendelian randomization, potentially causal gene expression changes for retinopathy that avoid potential confounding factors such as duration of diabetes and level of glycemia.

Mendelian randomization provides support for causality in this manuscript. We agree with the reviewer’s point that further corroborating functional evidence from the standpoint of molecular and cell biological assays in vitro and in vivo is lacking. We have changed the language throughout the manuscript to reflect this point. We have also amended the title accordingly to “Integration of genomics and transcriptomics predicts diabetic retinopathy susceptibility genes”.

Overall, the small group of studied subjects present huge differences in duration of diabetes and glucose control the 2 main factors of retinopathy. Thus, it is unclear how to differentiate the biological effects of long term high glucose and their impact on retinopathy.

As expected, differences existed between the cases and controls used in the gene expression studies in the primary risk factors for the development of complications including the level of glycemia and duration of diabetes. As the reviewer notes, it would be unclear how to distinguish fundamentally the role of any identified gene expression differences in this situation as being potentially causal for the development of retinopathy as opposed to purely being an epiphenomenon or, potentially worse, a change that is due to reverse causality from the disease itself (e.g. elevated HbA1c on the gene expression profile). Therein, in our opinion, lies a major strength of this paper through its incorporation of a completely independent, orthogonal approach of genomic analysis. By itself, the gene expression study provides only an association which could be significantly confounded or biased from uncontrolled covariates. The strength of genetic analysis in this setting is that a DNA nucleotide and its association to a disease cannot be confounded by duration of diabetes or the level of glycemia. While gene expression can be altered by both of these covariates, the genome sequence cannot.

To control for potential confounding, a paired analysis compared gene expression between individuals with and without retinopathy who were matched based on age, sex, treatment group, cohort and diabetes duration. Due to the dominant role in the development of retinopathy played by glycemia, we were not able to match individuals between the two groups based on this covariate. The genetic analyses did however account for differences in level of glycemia by controlling for HbA1c. Likewise, given the key role that duration of diabetes plays in the development of retinopathy, this factor also was not able to be completely controlled through matching between the two groups for the gene expression experiments.

In response to the reviewer’s comment, we revised the text of the manuscript further emphasizing (particularly in the Materials and methods and the Results sections) the use of a paired study design with individuals in the gene expression studies matched for age, sex, treatment group cohort, and diabetes duration. For example, we specifically made the following revisions to the manuscript text:

a) We added the following to the description of matching in the “Cell Lines” subsection of the Materials and methods: “For the seven pairs matched on duration: 4 pairs were matched by duration quartiles (baseline duration 0-4 years, 4-8 years, 8-12 years, or 12-15 years) and 3 pairs were matched by duration halves (<8 years vs. >=8 years). Matching by age was done similarly to duration: four pairs by quartile (<21 years, 21-25 years, 26-31 years, >=31 years) and the remaining four by halves (<26 years vs. >=26 years).”

b) We revised the sentence “Differential expression is described using fold change (FC) while differential response reflects fold change (FC) differences between groups” to “Differential response reflects fold change (FC) differences between matched individuals in the two groups in their paired response to glucose”.

c) We have changed the sentence “As anticipated, notable differences were observed between individuals with and without retinopathy (PDR vs. nDR) for duration of diabetes (53 +/- 43.4 months vs. 27 +/- 13.4 months) and mean HbA1c (9.71 +/- 2.37 vs. 7.62 +/- 1.07), respectively, given their significant impact on retinopathy“ to “As anticipated, notable differences were observed between individuals with and without retinopathy (PDR vs. nDR) for mean duration of diabetes (53 ±43.4 months vs. 27 ± 13.4 months) as it was also not possible to completely match participant pairs for this covariate or for level of glycemia (HbA1c), mean HbA1c (9.71 ± 2.37 vs. 7.62 ± 1.07) given their significant impact on retinopathy”.

2) Based on the transcriptome analysis the conclusion "This finding suggests that chronic glucose exposure depresses cellular immune responsiveness and may explain in part the increased risk of infection found in patients with diabetes" is not based on evidence as authors selected transcripts of their choice and also causality is not shown."Individuals with diabetic retinopathy exhibit a differential transcriptional response to glucose". Note that the level of association shown (especially for PDGF) is somewhat marginal.

As noted by the reviewer, the transcriptome analysis is underpowered. We agree with the reviewer that the underpowered nature of the transcriptional analyses are insufficient in themselves, given the marginal levels of significance, to suggest more than a limited association, which is why we employed a completely independent orthogonal approach of genomic analysis to validate the gene expression findings. We have accordingly amended the manuscript so as to not overstate the significance of the gene expression findings:

We have changed the sentence “Conversely, genes that modulate the cellular response to infection were considerably down-regulated (type 1 Interferon, FDR < 0.0001; gamma Interferon, FDR < 0.0001; leukocyte chemotaxis genes, FDR < 0.0001) and may explain in part the increased risk of infection found in patients with diabetes…” to “Conversely, genes that modulate the cellular response to infection were considerably down-regulated (type 1 Interferon, FDR < 0.0001; gamma Interferon, FDR < 0.0001; leukocyte chemotaxis genes, FDR < 0.0001) potentially supporting earlier work that chronic glucose exposure depresses cellular immune responsiveness”.

The goal of the gene expression analysis was not to demonstrate causality, but rather as a cross-check to support the biological plausibility of the response to glucose gene expression assay. All validation was done through the genomic analyses. The preliminary gene expression studies were purely hypothesis-generating. Prior to undertaking the genomic analyses, we investigated whether assessing the gene expression response to glucose in lymphoblastoid cell lines provided biologically plausible findings. Assessment for biological plausibility was conducted at both the gene level, as well as the pathway level. The gene level analysis identified *TXNIP*, a highly glucose-inducible gene in multiple cell types. On the pathway level we performed gene set enrichment analyses and identified several pathways that were down-regulated at an FDR < or = 0.0001 including type 1 gamma interferon response and leukocyte chemotaxis. The identification of these pathways was done through an analysis of the entire transcriptome in a completely unbiased, agnostic fashion with GSEA. Associated pathways at an FDR threshold < or = 0.0001 were highlighted. We felt that the pathway analysis, in addition to the single-gene analysis, supported the biological plausibility of the lymphoblastoid cell line gene expression response to glucose assay, given the pre-existing literature support for depressed cellular responsiveness in diabetes.

3) "Genes with differential response to glucose are implicated in the pathogenesis of diabetic retinopathy." This part is the most intriguing and original but it is based on expression in many tissues and thus the title is also overstated: it shows some kind of association but certainly not that the 103 genes "are implicated" in retinopathy.

We agree that not all 103 genes are likely associated with retinopathy. We have changed the title of this subsection to “Genetic association reveals that some genes with differential response to glucose play a role in susceptibility to diabetic retinopathy” so as not to overstate the nature of the findings. We used eQTLs from multiple tissues from the GTEx Project, because eQTLs that are shared between tissues show stronger associations, specifically for complex-trait associations in the retina. Retina eQTLs are not currently present in GTEx, although 43% of retina eQTLs are shared with LCLs.

4) "Folliculin (FLCN) is a putative diabetic retinopathy disease gene" this part is also interesting (and includes some in vivo experiments) but the original whole genome gene expression study did not detect FLCN as differentially expressed in the cell blood of the patients with retinopathy. Can the authors comment on this?

In fact, the original whole genome expression study did detect differential expression of *FLCN* between individuals with proliferative diabetic retinopathy and individuals without diabetic retinopathy (log_2_FC difference = 0.27, P = 2.5x10^-3^). We agree with the reviewer that this point could be made clearer in the manuscript. Accordingly, we have added text to this section indicating that the differential *FLCN* expression can also be seen in Figure 2 and Supplementary file 1B of the manuscript in addition to Figure 4—figure supplement 2.

One of the reviewers also mentioned that no SNPs near FLCN have been identified in diabetes (or complications) GWAS and this is somewhat worrying.

In our earlier GWAS published in Human Molecular Genetics (Grassi M.A. et al., 2011) we identified an association with diabetic retinopathy for the SNP rs11867934. rs11867934 is an eSNP for *FLCN* that has shown an association with diabetic retinopathy in two independent cohorts: GoKinD (P=6.1 X 10^-4^), EDIC (P=3.3 X 10^-3^), and their meta-analysis (P= 7 X 10^-6^). In the present manuscript, Figure 4—figure supplement 3 and Figure 4B reveal an enrichment of small GWAS p-values associated for the eSNPs associated with *FLCN* expression. Figure 4—figure supplement 4 demonstrates replication of these findings in a third cohort, the UK Biobank. Together, these analyses show the validity and reproducibility of these findings in separate, independent large cohorts of individuals with diabetic retinopathy.

GWASes for diabetic retinopathy have been historically underpowered. Applying a gene-based eQTL approach that aggregates the effects of multiple variants to a single testing unit, the gene, increases study power to identify a novel disease-associated locus by reducing the multiple testing burden by at least two orders of magnitude. This has allowed us, in an innovative and novel fashion, to identify signals that have previously not been identified for diabetic retinopathy. Examining the genome-wide association signal for disease from eQTLs for a gene in aggregate can be a more powerful strategy to discern heterogeneous genetic signals than testing each of these SNPs individually. Collating all of the eSNPs for *FLCN*, we assessed the aggregate association of *FLCN* eSNPs to diabetic retinopathy and observed an enrichment with a true positive rate of 0.9 in the GoKinD EDIC meta-analysis and a true positive rate of 0.7 in the UK Biobank. Such an approach has never been done before for diabetic retinopathy and speaks to a strength of the study and the reason for the novelty of the findings.

5) It is confusing that the authors used different selection criteria for gene identifications. In Results (subsection “Individuals with diabetic retinopathy exhibit a differential transcriptional response to glucose”), they identified 19 differentially response genes (P <0.05) between retinopathy cases and controls. However, they have selected the top 103 genes with P<0.01 (Results, subsection “Genes with differential response to glucose are implicated in the pathogenesis of diabetic retinopathy”) for further investigations. What is the gap between these two gene sets selection? I assume that the FLCN gene is in the top 103 gene set but not in the above 19 gene set. Explanations are needed for including specific genes for different analysis purposes.

We agree with the reviewer. In order to address this point we have changed the volcano plot for Figure 2 to highlight all 103 genes (P < 0.01) (red dots) instead of the 19 originally highlighted and updated Supplementary file 1B to reflect this point by including a list of these 103 genes. Both Figure 2 and Supplementary file 1B include *FLCN*.

The 19 and 103 gene sets come from the same list of differential glucose response genes and rather reflect different parameters. One set simply reflects the 103 genes with P < 0.01, whereas the other set of 19 genes had P < 0.05 and a fold-change difference of > or =0.26 (thresholds which were solely chosen to highlight their position on the volcano plot as well as the pre-existing literature support for some of these genes in terms of their role in retinopathy and their biological plausibility). Both the 19 and 103 gene sets contain *FLCN* and were derived from the differential response to glucose gene expression analysis between individuals with and without retinopathy. The entire data set listing all of these genes can be found in the source files for the manuscript.

6) The authors selected LCLs from individuals of 3 groups, non-diabetes (nDM), type 1 diabetes without retinopathy (nDR) and type 1 diabetes with proliferative diabetic retinopathy (PDR). The benefits of utilizing nDM samples in the analysis was not clear. What is the purpose of including these samples?

We agree with the reviewer that the nDM individuals were not central to the main message of the study. We initially included the nDM individuals because prior to this experiment we did not know whether we would see any difference between the gene expression profiles of individuals with and without retinopathy and how this might differ simply between individuals with and without diabetes itself. Our earlier work (Grassi et al., 2016) that interrogated expression differences for a small subset of candidate genes in these groups suggested we might not see a difference between individuals with and without retinopathy. If a difference was indeed present in the individuals with retinopathy, it was also unclear whether that would be due to a result of the differences in diabetes severity (from the standpoint of known differences in levels of glycemia and duration of diabetes between individuals with and without retinopathy), or due to underlying genetic differences. For this reason, we decided it would be prudent to include a group with no diabetes (nDM) for comparison purposes. It was only after we performed the multidimensional scaling analysis in which we saw clustering based on gene expression, revealing differences in individuals with and without retinopathy (Figure 2—figure supplement 1), and saw that these differences were grounded in known biology of diabetes and complications (Figure 2—figure supplement 2), that we then narrowed our focus to disentangling and better discerning the genetic basis for these differences. Ultimately, we decided to keep the nDM samples in the manuscript because their inclusion was relevant to the first question of simply whether there is interindividual variation in the transcriptional response (RG_all_). Including the nDM individuals increased the power of the RG_all_ analysis by increasing the sample size substantially. All of the findings from the analysis of the nDM group itself and in comparison to the other two groups can be found in the source files. We felt that including these specific analyses for the nDM group in the manuscript distracted from the primary findings presented for retinopathy.

Although both gene expression and GSEA methods were conducted, the results were not relevant to diabetic retinopathy. Similarly, it is not clear what the purpose of using the gene set enrichment analysis (GSEA) was because it seems that the authors performed most analyses to identify genetic components by gene-based or SNP-based methods in the manuscript.

GSEA was performed to support the biological plausibility of assessing glucose response in LCLs. The initial analyses assessing the response to glucose in the entire cohort (RG_all_, which included nDM, nDR and PDR) investigated whether there was any inter-individual variation in the transcriptional response to glucose (Figure 1—figure supplements 1-4), and whether the genes showing an inter-individual response to glucose were biologically plausible for diabetes and diabetic complications at both gene-level (e.g. *TXNIP*) (Figure 1A) and pathway-level (e.g. PDGF) (Figure 1B). Including the nDM individuals in this portion of the analysis increased the power of the study to identify these changes by increasing the sample size almost 50%.

7) The authors tested gene expression profile and associations using data from type 1 diabetic retinopathy. However, for the confirmation with UK BioBank (UKBB) data, they included all samples with both type 1 and type 2 diabetes. Did you perform the analysis stratified by the type of diabetes? Do you have any explanations of possible differences?

We decided to study individuals with both type 1 and type 2 diabetes from the UKBB to increase the study power as has been done in other consortia in which we have participated (Pollack et al., 2019; Meng, W et al., Acta Ophthalmologica, 2018). In order to account for the discrepancy between analyzing type 1 individuals (in GoKinD and EDIC) and type 2 individuals (in UKBB), we controlled for type of diabetes by including it as a covariate in our model for the GWAS in the UK Biobank. In fact, controlling for type of diabetes diminished the power of the UKBB GWAS (TPR 0.76 without vs. 0.73 with type of diabetes as a covariate) likely explaining some its decreased true positive rate compared to the GoKinD/EDIC meta-analysis. We did not perform a stratified analysis based on type of diabetes as the vast majority of individuals in the UK Biobank have type 2 diabetes. For our analysis, likely over 90% of cases had type 2 diabetes (8% had type 1 diabetes, but 15% were unspecified – the vast majority of which one would assume is likely type 2 diabetes); for controls, this number was even more pronounced with only 3% having type 1 diabetes. Hence, almost all of our association signals were found in individuals with type 2 diabetes. We were not sufficiently powered to reliably assess this signal only in individuals with type 1 diabetes. From a clinical and biological standpoint, we were most interested in identifying genes that predispose to retinopathy in the setting of diabetes, whether it be type 1, type 2 or unspecified.